# A Retrospective Experience of *Helicobacter pylori* Histology in a Large Sample of Subjects in Northern Italy

**DOI:** 10.3390/life11070650

**Published:** 2021-07-04

**Authors:** Davide Giuseppe Ribaldone, Carlo Zurlo, Sharmila Fagoonee, Chiara Rosso, Angelo Armandi, Gian Paolo Caviglia, Giorgio Maria Saracco, Rinaldo Pellicano

**Affiliations:** 1Department of Medical Sciences, Division of Gastroenterology, University of Torino, 10124 Torino, Italy; carlozurlo88@gmail.com (C.Z.); chiara.rosso@unito.it (C.R.); angelo.armandi@unito.it (A.A.); giorgiomaria.saracco@unito.it (G.M.S.); 2Institute of Biostructure and Bioimaging (CNR), Molecular Biotechnology Center, 10126 Turin, Italy; sharmila.fagoonee@unito.it; 3Department of General and Specialist Medicine, Gastroenterologia-U, Città della Salute e della Scienza di Torino, C.so Bramante 88, 10126 Turin, Italy; rinaldo_pellican@hotmail.com

**Keywords:** fundus, body, corpus, antrum, duodenum

## Abstract

Updated data about the prevalence of *Helicobacter pylori* (*H. pylori*) and its correlation with histological results are scarce. The aim of our study was to provide current data on the impact of *H. pylori* in a third-level endoscopy service. We performed a large, retrospective study analyzing the results of all histological samples of gastroscopy from the year 2019. In total, 1512 subjects were included. The prevalence of *H. pylori* was 16.8%. A significant difference between the prevalence in subjects born in Italy and those from eastern Europe, south America, or Africa was found (*p* < 0.0001, *p* = 0.006, and *p* = 0.0006, respectively). An association was found between *H. pylori* and active superficial gastritis (*p* < 0.0001). Current *H. pylori* and/or a previous finding of *H. pylori* was related to antral atrophy (*p* < 0.0001). Fifteen patients had low-grade dysplasia. There were no statistically significant associations with current or past *H. pylori* infection. One patient presented gastric cardia adenocarcinoma with regular gastric mucosa. One patient, *H. pylori* positive, was diagnosed with gastric signet ring cell adenocarcinoma in a setting of diffuse atrophy, without metaplasia.. Our study provides updated, solid (biopsy diagnosis and large population) data on the prevalence of *H. pylori* infection in a representative region of southern Europe.

## 1. Introduction

*Helicobacter pylori* (*H. pylori*) infection is the most common chronic bacterial infection in the world and is associated with numerous gastro-duodenal diseases such as duodenal ulcer, gastric ulcers, gastritis, gastric adenocarcinoma, and gastric MALT (mucosa-associated lymphoid tissue) lymphoma [1]. The prevalence of *H. pylori* shows large geographical variations: in developing countries, about 80% of the population is *H. pylori*-positive, even at a young age while, in industrialized countries, it generally remains below 40%, and is considerably lower in children and adolescents than in adults and the elderly [2,3,4].

*H. pylori* infection is the most frequent cause of metaplastic and dysplastic changes in the gastric mucosa and, for this reason, it is classified as a type I carcinogenic factor [5]. The search for *H. pylori* infection in gastric biopsies during a gastroscopy is also the most frequently made request [6].

European guidelines recommend using the “test-and-treat” strategy for *H. pylori* infection in countries where the prevalence of *H. pylori* is greater than 20% [5]. The data in the literature indicate that, in Italy, as in other southern European countries (e.g., Spain and Greece), the prevalence of *H. pylori* in adults appears to be around 50% [7,8]. Therefore, the “test-and-treat” strategy recommended in Italy.

Obtaining updated data on the prevalence of *H. pylori* in subjects belonging to a high-level specialist centre is important, both in terms of epidemiological data and to modify the clinical management of patients with gastric symptoms.

The main objective of this study is therefore a retrospective evaluation of *Helicobacter pylori* histology in a large sample of subjects who used the endoscopy service of the “A.O.U. Città della Salute e della Scienza di Torino”, Italy. The secondary objective is to evaluate and correlate the infection with pathological conditions based on the location, the activity of gastric inflammation, and the localization and extent of intestinal atrophy and metaplasia.

## 2. Materials and Methods

This retrospective study was based on the analysis of all gastric biopsies of patients referred to the endoscopy service in the period from 2 January 2019 to 31 December 2019.

The inclusion criteria were:

- At least two biopsy samples in the antrum to search for *H. pylori* infection;

- Age > 16 years.

The exclusion criterion were:

- A repeated examination in the same patient during the year.

*H. pylori* was searched for in all patients with gastric biopsies using the Giemsa stain. The sensitivity of the Giemsa stain for the detection of *H. pylori* is about 93%, while its specificity is about 100% [9]. Proton pump inhibitors (PPI) are stopped at least two weeks before gastroscopy (this is to improve the accuracy of *H. pylori* detection in patients undergoing biopsies in the antrum only [10])

Using the hospital’s centralized clinical reporting and archiving system, and the pathological anatomy service’s archiving and reporting system, all patients who underwent gastric biopsies during endoscopy in 2019 were selected.

The following clinical data were collected: age, sex, place of birth, previous gastroscopies, previous *H. pylori* infection, indication for gastroscopy, endoscopic result, localization of biopsies, and presence of *H. pylori*.

The study was conducted according to the guidelines of the Declaration of Helsinki. Ethical review, and approval was waived for this study due to the retrospective and anonymous data collection design. The hospital provided the form “CONSENT TO DATA PROCESSING FOR EXPERIMENTAL PURPOSES”. The shared database was collected anonymously. All the patients signed informed consent prior to examination (gastroscopy with biopsies). The STROBE guidelines were followed.

The primary outcome was the prevalence of *H. pylori* infection. The secondary outcomes were the correlation of the infection with histological conditions based on the location, the activity of gastric inflammation, and the localization and extent of intestinal atrophy and metaplasia.

Based on the presence or absence of inflammatory infiltrate, gastric atrophy, and intestinal metaplasia, as well as the localization of these alterations (only in the antrum, only in body/fund, or both sites), it was decided to associate the presence of atrophy only, i.e., without chronic inflammatory infiltrate, with the definition of atrophic gastritis, while considering the distinction regarding the nature of the inflammatory infiltrate (active or inactive).

### Statistical Analysis

Continuous variables were reported as mean (standard deviation (SD)), geometric mean, or median (interquartile range (IQR), depending on data distribution. The normality of the data was evaluated by the D’Agostino–Pearson test. Categorical variables were summarized as frequency and percentage. The Chi-square, or Fisher’s exact test when appropriate, and the independent sample t-test or Mann–Whitney test were applied for categorical and continuous variables, respectively. Logistic regression (multivariate) analysis was performed, including all statistically significant parameters to the univariate analysis and using the “enter” method.

A *p* < 0.05 was considered statistically significant. The statistical analysis was performed with MedCalc Statistical Software version 18.9.1 (MedCalc Software bvba, Ostend, Belgium).

## 3. Results

During the inclusion period, 1540 patients who underwent gastroscopy with gastric biopsy were selected. Of these, 28 subjects were excluded due to inadequate histopathological examination results (ongoing PPI *n* = 23, one single biopsy specimens *n* = 4). In total, 1512 subjects were included in the final analysis. Table 1 shows the characteristics of the study population.

In total, 243 (16.1%) examinations showed normally endoscopic findings, 585 (38.7%) had antral hyperaemia or erosions in the stomach or the duodenum, 76 (5%) had single or multiple gastric and/or duodenal ulceration, and 544 (36%) showed atrophic gastric mucosa, while other conditions (64 patients, 4.2%) included single or multiple gastric polyps (37), congestive (2), oedematous (10), nodular (3), or gastropathy (or other, 12 patients).

In 1193 (78.9%) patients, multiple biopsies were performed in the antrum and in the body/fundus (staging according to Operative Link on Gastritis Assessment, OLGA Staging system). In 319 (21.1%) patients, random samples were collected in the antrum only. Histological results are reported in Table 2 and Table 3.

The prevalence of *H. pylori* infection in gastric biopsy was 254 out of 1512 (16.8%). The prevalence of *H. pylori* infection in patients at their first gastroscopy was 209 out of 890 (23.5%); in patients who had already undergone at least one gastroscopy, the prevalence of *H. pylori* infection was 45 out of 622 (7.2%) (*p* < 0.001). The prevalence of *H. pylori* infection in patients who had never undergone a gastroscopy, and with no known history of *H. pylori* infection, was 197 out of 804 (24.5%). The prevalence of *H. pylori* infection in patients with a previous diagnosis of infection was 30 out of 195 (15.4%). Table 4 summarizes the characteristics of the population and the prevalence of *H. pylori* infection.

There was no statistically significant correlation between the prevalence of *H. pylori* infection for patients born in southern, northern, or central Italy (there was a trend towards a higher prevalence for those born in southern Italy compared to those born in northern Italy, *p* = 0.067). There was a statistically significant difference (*p* < 0.0001) between the prevalence in subjects born in Italy and those born in eastern Europe. There was also a statistically significant difference between the prevalence in subjects born in Italy and those born in south America (*p* = 0.0061), as well as between subjects born in Italy and those born in Africa (*p* = 0.0006). Through multivariable regression analysis, geographical origin from south America or Africa remained statistically significant (*p* = 0.04 and *p* = 0.005, respectively). A significantly higher prevalence of *H. pylori* was found in follow-up gastroscopies for atrophic gastropathy or previous peptic ulcer (*p* < 0.0001), which was confirmed by multivariate analysis (*p* = 0.002).

The following prevalence of *H. pylori* infection was found in relation to endoscopic findings: normal 12.8% (31/243); atrophic gastropathy 14.5% (79/544); hyperaemia or erosions 19.5% (114/585) (*p* = 0.02 compared with normal or atrophic gastritis); and ulcers 31.6% (24/76) (*p* = 0.0002 compared with normal or atrophic gastritis).

The prevalence of *H. pylori* infection was 206 out of 1193 (17.2%) in the 1193 patients from whom antrum and body/fundus samples were taken. In the 319 patients who underwent antral-only biopsies, the prevalence of *H. pylori* infection was 46 out of 319 (15.7%). The histological results associated with *H. pylori* infection are reported in Table 5.

An association was found between *H. pylori* and active superficial gastritis of the antrum and body/fundus (*p* < 0.0001), which was confirmed by multivariate analysis (*p* = 0.003). By correlating the finding of current *H. pylori* and antral atrophy, the association was found to be at the limit of statistical significance (*p* = 0.068); considering a history of *H. pylori* (current *H. pylori* and/or previous finding of *H. pylori*), a significant correlation was observed (*p* < 0.0001), which was confirmed by multivariate analysis (*p* = 0.004). The presence of antral metaplasia was associated with antral atrophy in 99.3% (299/301) of the patients (*p* < 0.0001, *p* = 0.005 at multivariate analysis). Body/fundus localized metaplasia was associated with atrophy in 96% (98/102) of the patients (*p* < 0.0001). Fifteen patients had low-grade dysplasia—14 in the antrum and 1 in the body/fundus. One patient presented gastric cardia adenocarcinoma with regular gastric mucosa. One patient had gastric sign cell adenocarcinoma with diffuse atrophy, without the metaplasia associated with *H. pylori* positivity. There were no statistically significant associations with current or past *H. pylori*.

## 4. Discussion

In the population of our study, we found *H. pylori* infection in 16.8% of subjects. This retrospective experience of *Helicobacter pylori* histology reported an infection rate lower than the known Italian prevalence, although the studies present in the literature are small in number, are dated and, above all, have been conducted on serological data by measuring anti-*H. pylori* IgG antibodies that do not give reliable information on active infection [11]. The data of the association between *H. pylori* and age showed a different trend compared to that in the literature: among *H. pylori*-positive patients, median age was 58 years compared to 62 years in *H. pylori*-negative patients; although, it can be speculated that the elderly population was more likely to undergo an invasive/non-invasive search for *H. pylori* in their lifetime and thus to have eradicated the bacterium.

In line with the literature data, however, was the data of the association between *H. pylori* infection and geographical origin. In fact, the prevalence of *H. pylori* infection was higher in Africa (40%), in south America, in particular in Peru (34.6%), and in eastern Europe (32.5%). Differences were statistically significant, despite the lower population of the groups under examination compared to patients of Italian origin. This is explained by the different prevalence of infection that still exists between developing countries and western countries [12]. As for Italy, the highest prevalence of *H. pylori* infection was found in southern Italy (17.2%), with a difference very close to statistical significance (*p* = 0.067).

Among the endoscopic results, it was confirmed that hyperaemia or erosions and ulcers are those most associated with *H. pylori* infection (*p* = 0.02 and *p* = 0.0002, respectively).

In 2002, Rugge et al. defined the staging of gastric histological changes associated with *H. pylori* (OLGA), emphasizing the difficulty of detecting *H. pylori* infection in cases of extensive intestinal metaplasia or during acid-suppressive therapy with PPI. In such cases, *H. pylori* infection is suggested by the presence of an “active” infiltrate [13].

In our study, among the histological findings, those associated with *H. pylori* infection were the following: superficial antrum and body/fundus gastritis (48.5%), chronic atrophic antral gastritis with chronic superficial body/fundus gastritis (38.3%), atrophic pangastritis (11.7%), and superficial gastritis of the antrum with atrophic gastritis of the body/fundus (1.5%). It should be noted that, among all the histological results associated with *H. pylori*, none of them showed involvement of the antrum only or the body/fundus only. All of these histological pictures showed an activity, namely the presence of polymorphonuclear infiltrate in the context of the biopsied gastric mucosa (100%).

*H. pylori* infection is the leading cause of gastric atrophy, and antral atrophic gastritis has been observed to occur in approximately half of the population with *H. pylori* infection. It was confirmed that the pictures of gastric atrophy and metaplasia localized at the body/fundus are rare and, in any case, influenced by the age of the subject and by the greater permanence of the bacterium in the stomach. The percentage of *H. pylori*-positive patients with antral metaplasia was 22.5%. In a similar vein to antral atrophy, no statistically significant association was found between antral metaplasia and current *H. pylori* infection; however, when considering patients with active *H. pylori* infection together with patients with a history of pylori infection, this association was statistically significant (*p* < 0.001), confirming what is known from the literature. Antral metaplasia occurs in the Correa cascade in the context of atrophy [14] and, in our study, a statistically significant association was found between gastric atrophy, both antrum and body/fundus, and gastric metaplasia (*p* < 0.001). From an anatomopathological point of view, it was decided not to carry out the characterization of metaplasia (complete, incomplete), despite the fact that, in the literature, incomplete metaplasia is a risk factor for the progression and malignant transformation of the gastric mucosa. However, in the same guidelines for the management of precancerous lesions, the clinical utility of the characterization of metaplasia requires further study [15].

The main limitation of our study is the fact that the observational and retrospective design may be subject to selection bias and may result in missing data concerning additional risk factors for the study outcomes, which partially precluded the acquisition of pre-endoscopy data. However, all patients eligible for the study were carefully screened and the sample size of our study is large.

## 5. Conclusions

Despite the aforementioned limitations, this study, representing a retrospective experience of *H. pylori* histology in a large sample of subjects in northern Italy, is one of the few that reports updated data based on biopsy samples and correlated them with histological pictures. Prospective studies are mandatory to confirm whether the real prevalence of *H. pylori* in Italy has decreased under a threshold for which a “test-and-treat” strategy could be no longer valid.

## Figures and Tables

**Table 1 life-11-00650-t001:** Baseline characteristics.

Characteristics of the population (*n* = 1512)	
Sex (*n*, %)	
- Males	597 (39.5%)
- Females	915 (60.5%)
AGE (years; median, IQR)	62.0, 47–77
Geographical origin (*n*, %)	
- Northern Italy	804 (53.2%)
- Central Italy	38 (2.5%)
- Southern Italy	510 (33.7%)
- Central/Western Europe	13 (0.9%)
- Eastern Europe	77 (5.1%)
- Asia	19 (1.2%)
- Africa	25 (1.7%)
- South America	26 (1.7%)
Previous *H. pylori* infection (*n*, %)	195 (12.9%)
Gastroscopy (*n*, %)	
- Naive	890 (58.9%)
- Experienced	622 (41.1%)
Gastroscopy indications (*n*, %)	
- Dyspepsia	642 (42.5%)
- GERD	304 (20.1%)
- Surveillance of atrophic gastritis/previous peptic ulcer/gastric MALT lymphoma	193 (12.8%)
- Anaemia	105 (6.9%)
- Other (search for signs of portal hypertension, dysphagia, family history for gastric cancer, suspected celiac disease, pre-renal transplant evaluation, etc.)	268 (17.7%)

IQR = interquartile range; *H. pylori* = *Helicobacter pylori*; GERD = gastro-esophageal reflux disease.

**Table 2 life-11-00650-t002:** A total 1193 patients on whom biopsies were performed in the antrum and body/fundus.

Result	*n*, %
Normal	159 (13.3%)
Chronic atrophic gastritis	182 (15.3%) (antral only, 178, body/fundus only 4)
• active inflammatory infiltrate	23
• inactive inflammatory infiltrate	120
• no inflammatory infiltrate	39
Chronic atrophic antral gastritis with chronic superficial body/fundus gastritis	221 (18.5%)
• chronic inflammatory infiltrate without activity	109
• chronic inflammatory infiltrate with activity	112
Chronic atrophic body/fundus gastritis with chronic superficial gastritis in the antrum	18 (1.4%)
• chronic inflammatory infiltrate without activity	8
• chronic inflammatory infiltrate with activity	8
• no inflammatory infiltrate	2
Chronic atrophic gastritis of the antrum and body/fundus	131 (11%)
• chronic inflammatory infiltrate without activity	73
• chronic inflammatory infiltrate with activity	50
• no inflammatory infiltrate	8
Superficial antral or body/fundus gastritis	187 (15.7%)
• chronic inflammatory infiltrate without activity	167
• chronic inflammatory infiltrate with activity	20
Chronic superficial gastritis of the antrum and body/fundus	295 (24.7%)
• chronic inflammatory infiltrate without activity	161
• chronic inflammatory infiltrate with activity	134

*n*, number of patients.

**Table 3 life-11-00650-t003:** A total 319 patients on whom biopsies were performed only in the antrum.

Result	*n*, %
Normal	54 (16.9%)
Chronic superficial antral gastritis	151 (47.3%)
• chronic inflammatory infiltrate without activity	93
• chronic inflammatory infiltrate with activity	58
Indefinite chronic antral gastritis due to atrophy or metaplasia due to small sample size	9 (2.8%)
• chronic inflammatory infiltrate without activity	7
• chronic inflammatory infiltrate with activity	2
Chronic atrophic antral gastritis	94 (29.5%)
• chronic inflammatory infiltrate without activity	55
• chronic inflammatory infiltrate with activity	21
• no inflammatory infiltrate	18
Insufficient material	3 (0.9%)

*n*, number of patients.

**Table 4 life-11-00650-t004:** Characteristics of the population and *Helicobacter pylori* status (*n* = 1512).

Characteristics opulation	*n*, % Median, IQR	*n*, % in *H. pylori* + (*n* = 254) Median, IQR	*n*, % in *H. pylori* − (*n* = 1258) Median, IQR	*p* Value
Sex				
Males	597 (39.5)	99 (39.0)	498 (39.6)	
Females	915 (60.5)	155 (61.0)	760 (60.4)	0.91
Age	62, 47–77	58, 42–74	62, 46–78	0.06
Geographic origin				
Northern Italy	804 (53.2)	109 (42.9)	695 (55.2)	0.061
Central Italy	38 (2.5)	6 (2.4)	32 (2.5)	0.89
Southern Italy	510 (33.7)	88 (34.6)	422 (33.5)	0.81
Central/Western Europe	13 (0.9)	2 (0.8)	11 (0.9)	0.79
Eastern Europe	77 (5.1%)	25 (9.8)	52 (4.1)	<0.0001
Asia	19 (1.2%)	5 (2.0)	14 (1.1)	0.057
Africa	25 (1.7%)	10 (3.9)	15 (1.2)	0.0003
South America	26 (1.7%)	9 (3.5)	17 (1.4)	0.005
Previous *H. pylori* infection	195 (12.9)	30 (11.8)	165 (13.1)	0.47
Gastroscopy				
naïve	890 (58.9%)	209	681	
Experienced	622 (41.1%)	45	577	<0.001
Indication TO EGDS				
Dyspepsia	642 (42.5)	129 (50.8)	513 (40.8)	0.061
GERD	304 (20.1)	54 (21.3)	250 (19.9)	0.56
Surveillance in atrophic gastritis or previous peptic ulcer	193 (12.8)	7	186	0.60
Anaemia	105 (6.9)	19 (7.5)	86 (6.8)	0.82
Other	268 (17.7)	45 (17.7)	223 (17.7)	0.99

*n*, number of patients; IQR, interquartile range; *H. pylori*, *Helicobacter pylori*; GERD, gastro-oesophageal reflux disease.

**Table 5 life-11-00650-t005:** Histological results and *Helicobacter pylori* status (*n* = 1512).

Histological Results (Antrum and Corpus/Fundus)	*n* = 1193	*H. Pylori* +	*H. Pylori* −	*p* Values
		*n* = 206	*n* = 987	
- Normal	159	0	159	*p* < 0.0001
- Superficial antral or body/fundus gastritis	187	0	187	*p* < 0.0001
- Superficial antral and body/fundus gastritis	295	100	195	*p* < 0.0001
- Chronic atrophic antral or body/fundus gastritis	182	0	182	*p* < 0.0001
- Chronic atrophic antral gastritis with chronic superficial body/fundus gastritis	221	79	142	*p* < 0.0001
- Chronic atrophic gastritis of antrum and body/fundus	131	24	107	*p* = 0.75
- Body/fundus atrophic gastritis with superficial antral gastritis	18	3	15	*p* = 0.95
**Histological results** **(antrum)**	***n*** **= 319**	***H. pylori*** **+**	***H. pylori*** **−**	
		***n*** **=** **48**	***n*** **= 271**	
- Normal	57	0	57	*p* < 0.0015
- Chronic superficial antral gastritis	151	31	120	*p* = 0.06
- Small sample size	9	1	8	*p* = 0.74
- Chronic atrophic antral gastritis	102	0	86	*p* = 0.0001

*n*, number of patients; *H. pylori*, *Helicobacter pylori*.

## Data Availability

The data were collected anonymously. Anonymous data can be requested in case of need.

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
