# Peer review of "A Retrospective Experience of Helicobacter pylori Histology in a Large Sample of Subjects in Northern Italy"

_life, 2021, doi:10.3390/life11070650_

Round 1

Reviewer 1 Report

This article reports the latest prevalence of H. pylori in Italy. It is noteworthy that the prevalence in Italy is lower than previously expected. On the other hand, the association between endoscopic findings and H. pylori infection was in good agreement with previous findings. The data collection and analysis methods of this study and the conclusions drawn from them are valid. This reviewer requests the authors to consider additional explanation on the following points.   Did all the determinations of H. pylori infections use biopsied tissue, and if so, what method was used? The diagnostic rate varies depending on the detection method, such as urea breath test (which does not require biopsy), RUT, culture, and microscopy (using biopsied tissue). If there is a mix of different diagnostic methods, the authors need to evaluate that the statistical significance is not dependent on those methods.

Author Response

Dear reviewer,

Thank you very much for appreciating our paper.

Q1) Did all the determinations of H. pylori infections use biopsied tissue, and if so, what method was used?

A1) We confirm that all the determinations of H. pylori infections were performed by tissue biopsy  using the Giemsa stain.

Reviewer 2 Report

I really doubt that a retrospective analysis may allow the prevalence of HP infection  to be determined. Since this is the primary aim of authors, I wonder what the interest of this study might be. Having said this, Authors may rephase the aim of the study (and related title) simply giving their retrospective experience of Hp histology in a large sample of subjects 

Author Response

Dear Auditor,

We understand your criticisms about the retrospective nature of the study. We point out that, despite this weakness, the study included all biopsies performed during a full year in a large center. Consequently, this study can reliably provide the prevalence of H. pylori infection when a patient undergoes gastroscopy, obtained by analyzing biopsies in more than 1500 in patients who stopped PPIs at least 2 weeks earlier.

We rephased the aim of the study (and related title) simply giving our retrospective experience of Hp histology in a large sample of subjects, according to your suggestions.

Round 2

Reviewer 1 Report

I request the authors to describe the H. pylori detection method (the Giemsa stain) and its expected average diagnostic rate based on literary reference(s) in the Materials and Method section.

Author Response

Dear Auditor,

Thanks again for your change request.

In the new version of the manuscript we have better specified that H. pylori was searched for in all patients with Giesma stain and we added a reference on specificity and sensitivity.